# TASKOGRAPHY: Evaluating robot task planning over large 3D scene graphs

**Project page: https://taskography.github.io**

Christopher Agia*[1], Krishna Murthy Jatavallabhula*[2], Mohamed Khodeir[1], Ondrej Miksik[3], Vibhav Vineet[3], Mustafa Mukadam[4], Liam Paull[2], and Florian Shkurti[1,5]

[1]University of Toronto, [2]Montreal Robotics and Embodied AI Lab, Mila, Université de Montréal, [3]Microsoft, [4]Facebook AI Research, [5]Vector Institute

**Abstract:** 3D scene graphs (3DSGs) are an emerging description; unifying symbolic, topological, and metric scene representations. However, typical 3DSGs contain hundreds of objects and symbols even for small environments; rendering task planning on the *full* graph impractical. We construct **TASKOGRAPHY**, the first large-scale robotic task planning benchmark over 3DSGs. While most benchmarking efforts in this area focus on *vision-based planning*, we systematically study *symbolic* planning, to decouple planning performance from visual representation learning. We observe that, among existing methods, neither classical nor learning-based planners are capable of real-time planning over *full* 3DSGs. Enabling real-time planning demands progress on *both* (a) sparsifying 3DSGs for tractable planning and (b) designing planners that better exploit 3DSG hierarchies. Towards the former goal, we propose SCRUB, a task-conditioned 3DSG sparsification method; enabling classical planners to match and in some cases surpass state-of-the-art learning-based planners. Towards the latter goal, we propose SEEK, a procedure enabling learning-based planners to exploit 3DSG structure, reducing the number of replanning queries required by current best approaches by an order of magnitude. We will open-source all code and baselines to spur further research along the intersections of robot task planning, learning and 3DSGs.

**Keywords:** Robot task planning, 3D scene graphs, learning to plan, benchmarks

## 1 Introduction

Real-world robotic task planning problems in large environments require reasoning over tens of thousands of object-action pairs. Faced with long-horizon tasks and an abundance of choices, state-of-the-art task planners struggle with an efficiency-reliability trade-off in grounding actions towards the goal. Hence, designing actionable scene abstractions suitable for a range of robotic tasks has drawn long-standing attention from the robotics and computer vision communities [1, 2, 3, 4, 5, 6].

A promising approach for building symbolic abstractions from raw perception data are 3D scene graphs (3DSGs, see Fig. 1) [7, 8, 9] – hierarchical representations of a scene that capture metric, semantic, and relational information, such as affordances, properties, and relationships among scene entities. While 3DSGs have to date been applied to simpler planning problems like goal-directed navigation [6, 10], active object search [11], and node classification [12], their amenability to more complex robotic task planning problems has yet to be thoroughly evaluated.

To investigate the joint application of 3DSGs and modern task planners to complex robotics tasks we propose **TASKOGRAPHY**: the first large-scale benchmark comprising a number of challenging task planning domains designed for 3DSGs. Analyzing planning times and costs on a diversity of domains in **TASKOGRAPHY** reveals that neither classical nor learning-based planners are capable of real-time planning over full 3DSGs, however, that they become so only when 3DSGs are sparsified.

Many real-world problems only require reasoning over a small subset of scene objects. E.g., the task "*fetch a mug from the kitchen*" primarily involves reasoning about scene elements associated with mugs or kitchens, rendering a vast majority of the remaining environment contextually irrelevant. Most planners aren't able to exploit such implicitly defined task contexts, instead spending most of their computation time reasoning about extraneous scene attributes and actions [13] (see Fig. 5).

---

*Authors contributed equally. Order determined by academic juniority.

5th Conference on Robot Learning (CoRL 2021), London, UK.

We argue that performant task planning over 3DSGs demands progress on two fronts: (a) sparsifying 3DSGs to make planning problems tractable, and (b) designing task planners that exploit the spatial hierarchies encapsulated in 3DSGs. To address (a), we present SCRUB–a planner-agnostic strategy guaranteed to produce a minimal *sufficient* object set for grounded planning problems. That is, planning on this reduced subset of scene entities suffices to solve the planning problem defined over the full 3DSG. Classical planning over state spaces (3DSGs) augmented by SCRUB outperforms state-of-the-art learning-based planners on the majority of tasks on our benchmark, without requiring any prior learning, establishing a strong baseline for future work in robotic task planning. To address (b), we present SEEK: a procedure tailored to 3DSGs, which supplements learning-based incremental planners by imposing 3DSG structure, ensuring all objects in the *sufficient* set are reachable from the start state. In our experiments, augmenting state-of-the-art planners with SEEK results in computational savings and an order of magnitude fewer replanning iterations.

In summary, we make the following contributions:

- TASKOGRAPHY: a large-scale benchmark to evaluate robotic task planning over 3DSGs,
- SCRUB: a planner-agnostic strategy to adapt 3DSGs for performant planning,
- SEEK: a procedure that enables learning-based planners to better exploit 3DSGs

We will open-source all code and baselines in TASKOGRAPHY-API, enabling the construction of new task planning domains, and benchmarking the performance of newer learning-based planners.

## 2    Related work

Early research in **symbolic planning** was centered around *optimal* planning [14, 15, 16, 17, 18]; planners producing solutions that preserve cost or plan length optimality. These methods are computationally expensive and thereby untenable to even moderately sized problems. This spurred work on *satisficing* planners that forgo optimal solutions for cheaper, feasible plans. Notable paradigms include regression planning [19], tree search [20], and heuristic search [21, 22, 23, 24, 25]. Whilst the many successes of heuristic planners [26, 27], computing low-cost informative heuristics is deterred by many extraneous objects [28, 13]; an inauspicious characteristic of large 3DSGs.

**Robot task planning** techniques have focused on constructing more effective representations to plan upon [29, 30, 31]. There are also approaches that integrate task and motion planning [32, 33, 34]–further demonstrated in hierarchical task space [35]–but which fall outside the scope of our work. Several approaches exploit task hierarchies for robot task planning [36] and control [37, 38, 39]. Different from these, our work focuses on exploiting abstractions in *spatial structure* encapsulated in 3DSGs, not to be conflated with hierarchical planners that exploit *task structure* [40].

State-of-the-art **learning-based** planners have demonstrated promising performance in small-to-moderate problem sizes. However, techniques such as relational policy learning [41], relational heuristic learning [42], action grounding [43], program guided symbolic planning [44, 45, 46, 47, 48], and regression planning networks [49] fail in large problem instances with branching factors and operators of the order considered (see Fig. 2) in the TASKOGRAPHY benchmark. Moreover, several planners that learn to search [50, 51, 52, 53] depend on hard-to-obtain dense rewards or do not scale with domain complexity [54, 55, 56].

The simplification of planning problems via **pruning strategies** to enable efficient search has been explored in both propositional [57, 58, 43] and numeric [59] planning contexts. Among these, PLOI [13] is a particularly performant learning-based approach that leverages object-centric relational reasoning [60, 61, 62, 63] to score and prune *extraneous objects* to the task. While PLOI outperforms existing classical planners on the TASKOGRAPHY benchmark, it incurs a large number of replanning steps owing to inaccurate neural network predictions; and inability to exploit 3DSG hierarchies. Our proposed SEEK procedure decreases replanning steps by two orders of magnitude.

**Planning benchmarks** in the symbolic planning communities have featured a variety of tasks with time complexities ranging from polynomial (e.g., shortest-path) to NP-hard problems (e.g., traveling salesman). There also exists a handful of environments [64, 65, 66, 67, 68] for benchmarking learned action policies from language directives and ego-centric visual observations, task and motion planning [69], or the modelling of physical interactions [70, 71]. Another recent benchmark [72] only supports navigation and block-stacking tasks. However, there isn't currently a large-scale benchmark tailored to robotic task planning in 3DSGs with several hundreds of objects.

# 3 Background

**Task planning.** A task planning problem $\Pi$ is a tuple $\langle \mathcal{O}, \mathcal{P}, \mathcal{A}, \mathcal{T}, \mathcal{C}, \mathcal{I}, \mathcal{G} \rangle$. As a running example, consider the task find an apple, slice it, and place it on the counter. $\mathcal{O}$ is the set of all ground objects (instances) in the problem. $\mathcal{P}$ is a set of properties, each defined over one or more objects; weight(apple) = 70 grams. **Predicates** are subclasses of properties in that they are boolean-valued; canPlace(apple, refrigerator) = True. $\mathcal{A}$ is a finite set of lifted actions operating over object tuples; slice(apple), place(apple, counter). $\mathcal{T}$ is a transition model and $\mathcal{C}$ denotes state transition costs. $\mathcal{I}$ and $\mathcal{G}$ are initial and goal states. A state is an assignment of values to all possible properties grounded over objects. For the running example, a goal state may be specified as on(apple, counter)=True and sliced(apple)=True. Planning problems may be grounded–slice *this* apple, or lifted–slice *an* apple.

**3D scene graphs (3DSGs).** A 3DSG [7, 8] is a hierarchical multigraph $G = (V, E)$ with $k \in \{1 \cdots K\}$ levels, where $V^k \in V$ denotes the set of vertices at level $k$. Edges originating from a vertex $v \in V^k$ may only terminate in $V^{k-1} \cup V^k \cup V^{k+1}$ (i.e., edges connect nodes within one level of each other). Each 3DSG in our work comprises at least 5 levels with increasing spatial precision as we move down the hierarchy: the topmost level in the hierarchy is a root node representing a scene. This node branches out to indicate the various *floors* in the building, which in turn branches out to denote various *rooms* in a floor, and subsequently *places* within a room. A place is a collection of *objects*, which may themselves contain other *objects* (to allow for container types such as cabinets and refrigerators).[2] At each level, edges indicate various types of relations among nodes (e.g., at the room level, an edge indicates the existence of a traversable path between two

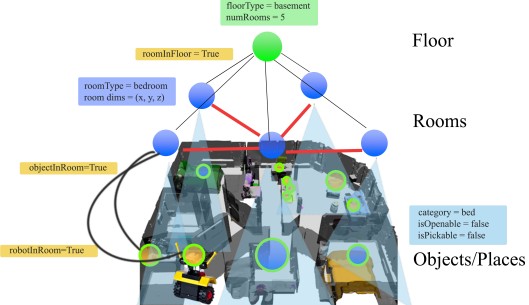

Figure 1: A *state* in a planning problem specified over a 3DSG. Nodes are scene entities and store unary predicates. Edges indicate binary predicates (relations). A goal is a conjunction of unary and binary literals. We only show a subset of relations for brevity. E.g., if the robot executes an action that moves it to another room, the robotInRoom relation shown in this figure will be set to False for the room on the lower left.

rooms; at the object level, edges indicate multiple affordance relations). Each node also stores semantic attributes such as node type, functionality, affordances, etc., following [7].

# 4 TASKOGRAPHY

We propose TASKOGRAPHY: the first large scale benchmark to evaluate symbolic planning over 3DSGs. Currently, TASKOGRAPHY comprises 20 challenging robotic task planning domains totaling 3734 tasks. Different from current benchmarks for embodied AI that focus primarily on egocentric *visual* reasoning [65, 73, 74, 64, 71, 67]; TASKOGRAPHY is designed to evaluate *symbolic* reasoning over 3DSGs. To emulate the complexity of real-world task planning problems, TASKOGRAPHY builds atop the Gibson [75] dataset comprising real-world scans of large building interiors (averaging 2-3 floors per building; 7 rooms per floor), and their corresponding 3DSGs [7].

**Augmenting 3DSGs with plannable attributes.** A prerequisite for planning over 3DSGs—absent in existing work [9, 7, 8]—is a database of *plannable attributes*: predicates, actions, and transition models. To support task planning, we augment each 3DSG in Gibson [75] (tiny and medium splits) with several layers of additional unary and binary predicates. For each 3DSG node, we obtain class labels, object dimensions and pose from [7]. We annotate object affordances by building a knowledge base of lifted object-action pairs and recursively applying it to every 3DSG node, while accounting for exceptions (objects that are concealed or contained within others). We also detect *door* objects in the 3DSG and use this to add additional edges describing room connectivity. We annotate objects with all possible properties in our planning domains (e.g., "*is this object typically a receptacle?*"). Our rich property set (*plannable attributes*) is chosen to support a wide range of realistic-robotic tasks geared towards large (building-scale) 3DSGs.

---

[2]The lowest level in [8] is a metric-semantic mesh. However since our focus is on symbolic planning, we only require scene graph levels that contain *objects*.

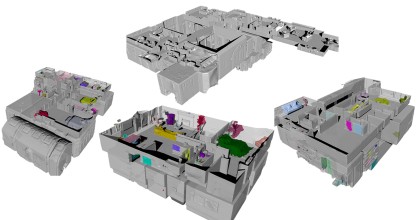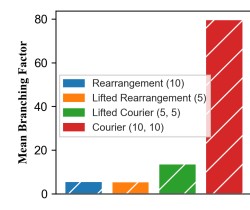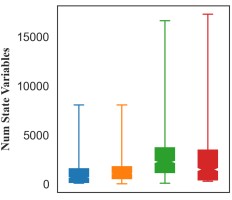

Figure 2: The **TASKOGRAPHY benchmark** comprises large-scale planning problems defined over buildings from the Gibson dataset [75]. (Left) Representative buildings from Gibson [75]. (Middle/Right) We feature a variety of problem classes ranging in scale and complexity as illustrated by the domain statistics.

**Benchmark statistics.** Each of the 20 TASKOGRAPHY domains specifies a class of planning problems that resemble real-world use cases (and theoretically complex extensions) that a robot would encounter in office, house, or building scale environments. These domains range from grounded planning domains to lifted planning domains, domains with no extraneous objects to domains where most objects are extraneous, and domains for which polynomial time solutions exist to NP-hard problems. The simplest domains in the benchmark have 1000 state variables and an average branching factor of 5; for hard domains, these are 4000 and 60 respectively (see Fig. 2).

**TASKOGRAPHY-API.** Our project page (https://taskography.github.io) will host code and data used in this work. In TASKOGRAPHY-API, an open-source python package, we provide access to 18 classical and learning-based symbolic planners, templates to implement novel domains, and methods to generate problem instances of varying complexities and train/evaluate learning-based planners.

**Planners considered.** TASKOGRAPHY supports a comprehensive set of planners to facilitate standardized evaluation on novel domains. The following planners are available at the time of writing.

- **Optimal planners**: Fast Downward (FD) with the `opt-lmcut` heuristic [23], Sat-Plan [16], Delfi [17], DecStar-optimal [25], and Monte Carlo tree search.
- **Satisficing planners**: Fast Forward (FF), FF with axioms (FF-X) [22], Fast Downward (FD) with the `lama-cut` heuristic [23], DecStar-satisficing [25], Cerberus [24], Best First Width Search (BFWS) [76], and regression planning.
- **Learning-based planners**: Relational policy learning [41], Planning with learned object importance (PLOI) [13] (and variants – see Sec. 6).

**General assumptions.** To facilitate evaluation of all of these classes of planners, the first edition of our benchmark only considers *fully observable* tasks and *discrete* state and action spaces. All goal states are specified as *conjunctions* of literals. While we make no distinction between deterministic or stochastic transitions, all current experiments assume a *closed world*, i.e., all possible lifted actions and effects are known apriori.

### 4.1 Robot planning domains: Case studies

The full TASKOGRAPHY benchmark comprises 20 domains. We discuss the four task categories from which all domains are constructed that we believe to be interesting to a broad robotics audience.

**Domain 1.** Rearrangement($k$)*: Based on the recently proposed rearrangement challenge [77], this task requires a robot randomly spawned to rearrange a set of $k$ objects of interest into $k$ corresponding receptacles. The robot often needs to execute multiple other actions along the way, such as opening/closing doors, navigating to goals, planning the sequence of objects to visit, etc.*

**Domain 2.** Courier $(n, k)$*: A robot that couriers objects is equipped with a knapsack of maximum payload capacity of $n$ units. The robot needs to locate and courier $k$ objects (of varying weights $w \in \{1, 2, 3\}$ units) to $k$ distinct delivery points. The knapsack can be used to stow and retrieve items in random-access fashion; effectively embedding a combinatorial optimization problem into the task. Stow and retrieve actions increase branching, necessitating far deeper searches.*

We also provide *lifted* variants of these tasks. Here, goals are specified over desired object-receptacle class relations (e.g., "put a cup on a table") as opposed to over object instances (e.g., "put this cup on the table"). These tasks introduce ambiguity in both the search of classical task-planners and learning-based techniques, which must now distinguish object instances of relevant classes.

Table 1: **TASKOGRAPHY** benchmark results on select grounded and lifted *Rearrangement* (**Rearr**) and *Courier* (**Cour**) 3DSG domains. Planning times are reported in seconds and do not incorporate planner-specific domain translation times (factored into planning timeouts). A '-' indicates planning timeouts or failures (10 minutes for optimal planners, 30 seconds for all others). Results are aggregated over 10 random seeds. Optimal task planning is infeasible in larger problem instances or for more complex domains, while most satisficing planners are unable to achieve real-time performance. PLOI [13], a recent learning-based planner consistently performs the best across all domains.

| | Planner | Rearr(1) Tiny | | | Rearr(2) Tiny | | | Rearr(10) Medium | | | Cour(7, 10) Medium | | | Lifted Rearr(5) Tiny | | | Lifted Cour(5, 5) Tiny | | |
|---|---|---|---|---|---|---|---|---|---|---|---|---|---|---|---|---|---|---|---|
| | | Len. | Time | Fail | Len. | Time | Fail | Len. | Time | Fail | Len. | Time | Fail | Len. | Time | Fail | Len. | Time | Fail |
| optimal | FD-seq-opt-lmcut | 15.77 | 24.81 | 0.04 | **25.80** | 104.47 | 0.55 | - | - | 1.00 | - | - | 1.00 | - | - | 1.00 | - | - | 1.00 |
| | SatPlan | 14.77 | 10.35 | 0.45 | 26.67 | 3.27 | 0.67 | - | - | 1.00 | - | - | 1.00 | - | - | 1.00 | - | - | 1.00 |
| | Delfi | 15.13 | 0.36 | 0.16 | 29.10 | 27.77 | 0.29 | - | - | 1.00 | - | - | 1.00 | - | - | 1.00 | - | - | 1.00 |
| | DecStar-opt-fb | - | - | 1.00 | - | - | 1.00 | - | - | 1.00 | - | - | 1.00 | - | - | 1.00 | - | - | 1.00 |
| | MCTS | - | - | 1.00 | - | - | 1.00 | - | - | 1.00 | - | - | 1.00 | - | - | 1.00 | - | - | 1.00 |
| satisficing | FF | 16.71 | 0.19 | **0.00** | 34.44 | 0.55 | **0.00** | 159.04 | 5.30 | 0.09 | 128.41 | 6.62 | 0.24 | 62.86 | 3.40 | 0.47 | **57.74** | 4.03 | 0.44 |
| | FF-X | 16.71 | 0.25 | **0.00** | 34.44 | 0.58 | **0.00** | 159.80 | 5.02 | 0.08 | 128.19 | 6.72 | 0.24 | 67.88 | 3.48 | 0.89 | 61.19 | 7.56 | 0.77 |
| | FD-lama-first | 15.19 | 2.96 | 0.33 | 38.47 | 3.25 | 0.18 | 208.28 | 6.35 | 0.49 | 156.34 | 4.92 | 0.29 | 66.81 | 3.20 | 0.49 | 61.13 | 3.34 | 0.56 |
| | Cerberus-sat | **11.50** | 12.00 | 0.85 | - | - | 1.00 | - | - | 1.00 | - | - | 1.00 | - | - | 1.00 | - | - | 1.00 |
| | Cerberus-agl | 14.77 | 5.13 | 0.45 | 33.00 | 7.30 | 0.49 | 176.60 | 8.91 | 0.72 | **125.73** | 12.99 | 0.83 | 60.50 | 7.62 | 0.60 | 59.19 | 7.05 | 0.77 |
| | DecStar-agl-fb | 14.72 | 2.62 | 0.55 | 34.96 | 2.58 | 0.58 | 211.16 | 7.20 | 0.82 | 132.60 | 4.50 | 0.58 | 66.30 | 3.02 | 0.71 | 58.75 | 4.46 | 0.71 |
| | BFWS | 15.56 | 0.90 | 0.22 | 32.16 | 0.37 | 0.18 | **151.17** | 0.41 | 0.23 | 152.71 | 1.13 | 0.21 | **56.90** | 0.94 | 0.41 | 61.92 | 2.30 | 0.43 |
| | Regression-plan | - | - | 1.00 | - | - | 1.00 | - | - | 1.00 | - | - | 1.00 | - | - | 1.00 | - | - | 1.00 |
| learn | Relational policy [41] | - | - | 1.00 | - | - | 1.00 | - | - | 1.00 | - | - | 1.00 | - | - | 1.00 | - | - | 1.00 |
| | PLOI [13] | 16.45 | **0.00*** | **0.00** | 37.04 | **0.00*** | **0.00** | 213.43 | **0.17** | **0.00** | 161.90 | **0.34** | **0.00** | 78.68 | **0.22** | **0.24** | 71.71 | **0.26** | **0.26** |

**Domain 3.** Lifted Rearrangement ($k$)*: A lifted version of the rearrangement domain where the goals are specified at an object category level, as opposed to an instance level.*

**Domain 4.** Lifted Courier ($n, k$)*: A lifted version of the courier domain where the goals are specified at an object category level, as opposed to an instance level.*

To promote compatability with a range of planning systems [27, 78], we represent all tasks in PDDL format [79, 80]. We also include mechanisms for translating tasks into alternative problem definition languages that are essential for some of our supported planners [16].

## 4.2 Benchmarking classical and learned planners on TASKOGRAPHY

We present the empirical results on the TASKOGRAPHY benchmark across several classes of task-planners in Table. 1. (Please consult supplementary material for a number of additional results).

**Evaluation protocol.** We treat the evaluation of optimal planners separately to the remaining methods. Optimal planners are not intended to be fast unlike satisficing and learning-based variants. Rather, they compute a solution of minimum length (not necessarily unique) to a given problem. Optimal planners are hence allotted 10 minutes to solve each problem, while satisficing and learning-based planners are allotted 30 seconds. For learning-based methods, we evaluate results over 10 random seeds for statistical significance. We report standard deviations in the supplementary material. All domains comprise 40 training problems. The domains tagged *Tiny* and *Medium* comprise 55 and 182 test problems respectively, unless otherwise specified.

**Optimal planners work only on the simplest of domains.** Despite the reasonable performance of optimal planners on the *Rearrangment(1)* domain, they are unable to efficiently scale with increasing task complexity and fail to solve a single task on the Rearrangment (k) and Courier (n, k) domains for $k > 2$. In particular, the Rearrangement(1) domain is a superset of the grounded hierarchical path planning (HPP) task as described by Rosinol et al. [6]. Because the HPP task does not consider state changes to the scene graph (i.e., directly equating the 3DSG to the planning graph for search), efficient shortest path planning is tractable. However, increasingly complex robot tasks requires more than the mere ability to path plan in 3DSGs.

**Planning performance degrades with domain complexity, not scene complexity.** We observe an increase in the number of planning failures and timeouts as satisficing planners are applied to larger *Rearrangement(k)* domains (Table 2). Interestingly, larger scenes do not appear to directly correlate with task complexity, as the performance metrics remain largely consistent between the tiny and medium splits of the same domain (Table 2).

Table 2: Interestingly, task complexity does not correlate strongly with scene complexity. It is instead determined by the number of operators, and avg. branch factor.

| Planner | Rearr(10) Tiny | | | Rearr(10) Medium | | |
|---|---|---|---|---|---|---|
| | Len. | Time | Fail | Len. | Time | Fail |
| FF | 162.61 | 7.04 | **0.07** | 159.04 | 5.30 | **0.09** |
| FD (satisficing) | 205.89 | 7.68 | 0.51 | 208.28 | 6.35 | 0.49 |
| DecStar-agl-fb | 193.00 | 6.78 | 0.85 | 211.16 | 7.20 | 0.82 |
| BFWS | **160.93** | 0.57 | 0.18 | **151.17** | **0.41** | 0.23 |

**Satisficing planners fail in domains requiring long-horizon reasoning.** In the *Courier(n, k)* domains, satisficing planners tend to produce shorter length solutions by leveraging the knapsack's capacity to stow objects on the way to various delivery points. However, the planners often display shortsighted behaviours by stowing objects early in the search, depleting knapsack slots that could potentially help further along the task. This yields dead-end configurations and excessive backtracking, and thus, an increase in timeouts is observed.

**Planners that do not exploit forward heuristics fail due to large branching factors.** Due to the large branching factor of our domains, common strategies such as Monte-Carlo Tree Search (MCTS) and MC Regression Planning are unable to solve any task within a reasonable time constraint. For instance, a *Rearrangement(10)* task has an average branching factor of 6.5 for MCTS. Since a reward is only obtained at the end (typical planners take 200 steps to get there), MCTS degenerates to a slow breadth-first search.

**Learning based planners that prune the state space excel on all domains.** We also evaluate two prevalent learning-to-plan methods based on generalized relational planning [41] and planning with learned abstractions [13]. While the relational policy stuggles to generalize in our domains (long-horizon, sparse rewards), PLOI demonstrates an impressive ability to detect and prune contextually irrelevant parts of the 3DSGs. However, it also requires a significant number of replanning steps (see figure to the right) as it often retains objects within a graph without ensuring that all properties and ancestors required to access the object are also preserved.

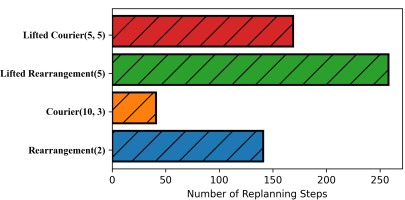

Figure 3: Learning-based planners like PLOI outperform all other planners on the benchmark, but still incur significant overhead (number of replanning steps).

**Discussion.** Our evaluation of existing performant planners on the TASKOGRAPHY benchmark consistently reveals two important trends across all domains.

- Pruning a 3DSG is essential for real-time performance, more so on challenging domains.
- While learning-based planners excel across all domains, they require a large number of replanning steps.

These imply that efficient utilization of 3DSGs in real-time robotic task planning requires *both* adapting 3DSGs to better suit existing planners, and enabling performant (learning-based) planners to better exploit 3DSG hierarchies. The remainder of our work addresses these issues.

## 5  SCRUB: Principled sparsification of 3DSGs for efficient planning

As discussed above, learning-based planners leverage a wealth of prior knowledge acquired during a training phase to significantly prune extraneous scene graph entities. We argue that, if equipped with the right sparsification machinery, classical planners can compete with, or outperform learning methods. We develop SCRUB, a principled 3DSG sparsification scheme that prunes a 3DSG $G$ (w.r.t. planning problem $\Pi_G = \langle \mathcal{O}, \mathcal{P}, \mathcal{A}, \mathcal{T}, \mathcal{C}, \mathcal{I}, \mathcal{G} \rangle$) by removing vertices and edges extraneous to the task, resulting in a sparsified 3DSG $\hat{G}$ (and planning problem $\hat{\Pi}_{\hat{G}} = \langle \hat{\mathcal{O}}, \hat{\mathcal{P}}, \hat{\mathcal{A}}, \hat{\mathcal{T}}, \hat{\mathcal{C}}, \hat{\mathcal{I}}, \mathcal{G} \rangle$)

**Definition 1.** *A valid* 3DSG *sparsification of $G$ for a planning problem $\Pi_G$ to $\hat{G}$ (and corresponding planning problem $\hat{\Pi}_{\hat{G}}$) is a computable function* SCRUB$(\Pi_G) = \hat{\Pi}_{\hat{G}}$ *such that, a plan $p$ solves $\Pi_g$ iff it solves $\hat{\Pi}_{\hat{G}}$.*

**Algorithm 1:** SCRUB

**Input:** 3DSG $G$, Planning problem
  $\Pi = \langle \mathcal{O}, \mathcal{P}, \mathcal{A}, \mathcal{T}, \mathcal{C}, \mathcal{I}, \mathcal{G} \rangle$
**Result:** Sparsified 3DSG $\hat{G}$
$\hat{\mathcal{O}} = \{\}$; /* Init. sufficient object set */
$g = $ OBJECTS$(\mathcal{G}$.literals$) \cup \{robot\}$; /* Init. set of objects in the goal literal set */
**while** *not empty g* **do**
  $\hat{\mathcal{O}} \leftarrow \hat{\mathcal{O}} \cup g$
  $p \leftarrow$ all binary predicates relating a newly added object (i.e. $o \in g - \hat{\mathcal{O}}$) to its ancestors in $G$
  $g \leftarrow$ OBJECTS$(p)$
  **if** *all objects $\mathcal{O}$ visited* **then**
    | break
  **end**
**end**
$\hat{G} \leftarrow G$; /* Initialize sparsified scenegraph */
CONNECTROOMS; /* All-pairs shortest paths */
Remove all nodes from $\hat{G}$ that are not in $\hat{\mathcal{O}}$
Prune literals that are no longer valid in the sparsified graph

A satisficing plan for $\Pi_G$ may thus be obtained by simply solving the (much easier to solve) sparsified problem $\hat{\Pi}_{\hat{G}}$. Savings in planning time depend on the complexity of the sparsified subgraph $\hat{G}$. SCRUB presents a simple strategy which is guaranteed to be minimal for grounded planning problems and satisficing for lifted planning problems.

For exposition, we consider grounded planning problems; see appendix for how SCRUB is adapted to lifted planning problems or stochastic transitions. SCRUB begins with an initially empty sufficient object set $\hat{\mathcal{O}}$. Satisfying the goal minimally requires all ground objects in the goal to be included in the sufficient object set $\hat{\mathcal{O}}$ (else goal objects are unreachable). In addition, the robot itself must be part of the sufficient set. Let $p$ be the set of all binary predicates which include any of these objects. And let $g$ be the set of all objects contained in $p$. In general, this will be a superset of the objects we started with. We iteratively repeat this process, each time adding the new objects in $g$ to our sufficient set $\hat{\mathcal{O}}$.

The process terminates either when either the set $g$ has no new objects (indicating convergence), or until all the objects in the scene graph are visited at least once (indicating the input graph already defines a minimal object set). We initialize the nodes of $\hat{G}$ with objects in $\hat{\mathcal{O}}$, and copy over all edges $(u, v) \in G$ for which both $u, v \in \hat{\mathcal{O}}$. SCRUB terminates in time linear in the number of the predicates or nodes (whichever is larger).

**Proposition 1.** SCRUB *is complete and results in a minimal scene subgraph for all grounded planning problems over the scenegraph domain. (Please refer to supp. material for proof)*

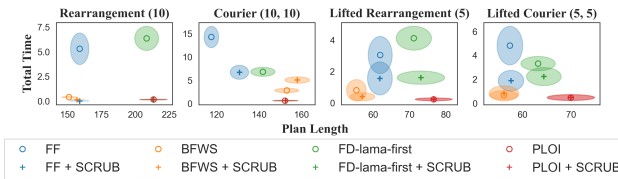

Figure 4: Best performing planners with and without SCRUB.

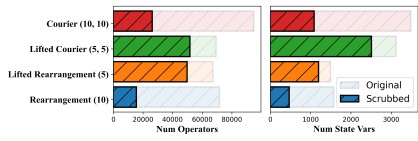

Figure 5: SCRUB greatly prunes operators and states of planning problems.

## 5.1 Impact of SCRUB on modern task planners

In this section, we investigate the effect that a 3DSG reduction scheme like SCRUB may have on the performance of modern task planners. We experiment with the four domains shown in Fig. 4 and evaluate the impact of scrub on planning performance and on domain structure.

**SCRUB enables classical planners to obtain performance at least as good as state-of-the-art planners.** In Fig. 4, we see that SCRUB drastically reduces planning time for `FF`, `FD-lama-first`, and `BFWS` to a few milliseconds on Rearrangement(10), and upper-bounds times at 5 seconds on Courier(10, 10). We see this enables BFWS, FD, and FF to outperform PLOI (lower plan lengths for similar plan times). The grounded domains each have 182 test problems, and the lifted domains each have 70 test problems.

**SCRUB greatly reduces the number of operators and states.** To asses the impact of SCRUB, we compute statistics (number of operators, number of state variables) in Fig. 5. We see that SCRUB prunes *more than two-thirds* of the operators and state variables for grounded planning problems, and about a third in the case of lifted planning problems.

**SCRUB enables optimal planners to run on lifted domains.** Table 3 reports results of running the satisficing and optimal variants of FD with and without SCRUB, on the *Lifted Rearrangement(5)* domain. While FD (optimal) did not converge even with a timeout of *24 hours*, FD (optimal) + scrub solved about 72% of the tasks under a 30-second timeout, taking 2 seconds per task on average.

Table 3: Planner statistics evaluated over 70 test problems on *Lifted Rearrangement*(5).

| Planner | % Success | Length | Time |
|---|---|---|---|
| **FD (satisficing)** | 51.43 | 66.81 | 3.20 |
| **FD (satisficing) + SCRUB** | 72.86 | 73.09 | 1.61 |
| **FD (optimal)** | - | - | - |
| **FD (optimal) + SCRUB** | 72.86 | 68.33 | 2.26 |

## 6 SEEK: A procedure for efficient learning-based planning

While SCRUB results in a 3DSG reduction that is guaranteed to find a satisficing plan—if one exists—its conservative approach hurts performance in challenging lifted planning problems as shown in Fig. 4. For such problems, learning-based graph-pruning strategies like PLOI [13] outperform classical planners over SCRUBBED 3DSGs. However, as can be seen in Sec. 4.2, even PLOI [13] incurs a significant number of replanning iterations.

Table 4: **SEEK** significantly reduces the number of replanning steps required by state-of-the-art learning-based planners. For each planner, we report average *wall time* (including translation time).

| Planner | Rearrangement (2) - Medium | | | | | Courier (10, 3) - Medium | | | | | Lifted Rearrangement (5) - Medium | | | | | Lifted Courier (5, 5) - Medium | | | | |
|---|---|---|---|---|---|---|---|---|---|---|---|---|---|---|---|---|---|---|---|---|
| | %Succ. | Len. | %Used | Time | #Replan | %Succ. | Len. | %Used | Time | #Replan | %Succ. | Len. | %Used | Time | #Replan | %Succ. | Len. | %Used | Time | #Replan |
| **Random** | 0.87 | 39.81 | 0.99 | 9.51 | 836 | 0.62 | 180 | 0.10 | 12.11 | 204 | 0.63 | 68.98 | 0.99 | 10.93 | 235 | 0.67 | 67.89 | 0.98 | 10.81 | 233 |
| **Random + SEEK** | 0.86 | 39.82 | 0.98 | 8.55 | **543** | 0.60 | 183.49 | 0.99 | 12.33 | **162** | 0.59 | 69.22 | 0.97 | 9.52 | **155** | 0.63 | 65.48 | 0.97 | 10.97 | **167** |
| **Hierarchical** | 1 | 35.76 | 0.28 | 0.45 | 150 | 1 | 191.75 | 0.48 | 1.16 | 40 | 0.80 | 76.75 | 0.59 | 2.60 | 269 | 0.73 | 69.69 | 0.61 | 2.73 | 173 |
| **Hierarchical + SEEK** | 1 | 35.76 | 0.28 | 0.30 | **12** | 1 | 191.75 | 0.48 | 0.97 | **7** | 0.80 | 76.70 | 0.56 | 2.20 | **208** | 0.77 | 76.04 | 0.55 | 1.59 | **76** |
| **PLOI [13]** | 1 | 35.76 | 0.28 | 0.44 | 141 | 1 | 191.75 | 0.48 | 1.13 | 41 | 0.79 | 78.16 | 0.59 | 2.49 | 258 | 0.73 | 69.88 | 0.62 | 2.75 | 169 |
| **PLOI + SEEK** | 1 | 35.76 | 0.28 | 0.31 | **14** | 1 | 191.75 | 0.48 | 0.97 | **7** | 0.80 | 76.61 | 0.56 | 2.18 | **197** | 0.77 | 79.19 | 0.55 | 1.53 | **53** |

We posit that several replanning iterations may be avoided by exploiting the 3DSG hierarchy. Pruning strategies like PLOI first score all objects, and retain a minimal set by thresholding. A simple threshold does little to ensure that all retained objects are reachable from the root of the scene graph. To alleviate this issue, we propose SEEK: a procedure that ensures we obtain a connected graph, with the objective of reducing the number of replanning steps needed.

SEEK requires as input the 3DSG, the planning problem $\Pi$, and an object scoring mechanism $f_\theta$. This scoring mechanism is typically a graph neural network (akin to [13]) that, given the current state, scores each object with an *importance* value in $[0, 1]$. We first run the scorer and only retain objects above a threshold score $t$. We follow an identical approach to PLOI [13] and at each step geometrically decay the threshold by $\gamma$, such that at iteration $i$, the threshold is $t_i = \gamma t_{i-1}$, with $t_0, \gamma \in [0, 1)$. For each retained object $o$, we recursively traverse up the 3DSG, adding all ancestors of $o$ to the sufficient object set. This procedure ensures that all objects are reachable from their respective room nodes. While SEEK, unlike SCRUB, is not guaranteed to be satisficing, it results in far fewer replanning steps without affecting computation time.

**SEEK reduces replanning steps by an order of magnitude**. To assess the impact of the SEEK procedure on planning performance, we evaluate performance with respect to other learning-based planners on TASKOGRAPHY in Table 4. As a baseline, we evaluate a *random* pruning strategy that uniformly randomly retains or prunes every object. Even for this naive strategy, SEEK offers significant performance improvement. We also evaluate *PLOI* [13] and our adaptation dubbed *hierarchical*, which trains multiple graph neural networks, one for each level of the 3DSG hierarchy. For each variant, SEEK offers a consistent performance improvement by decreasing the number of replanning steps required as seen in Fig. 6. SEEK is thus a conceptually simple strategy for use with learning-based planners.

**SCRUB on grounded domains, SEEK on lifted domains**: In general, we note that SCRUB is more performant on grounded domains (due to minimality properties) and SEEK is more performant on lifted domains (where SCRUB typically retains all instances of important object categories, but SEEK is more effective due to its opportunistic retention of instances (Fig. 7)).

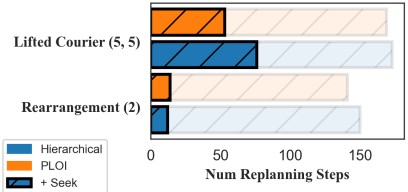

Figure 6: SEEK reduces replanning steps by an order of magnitude.

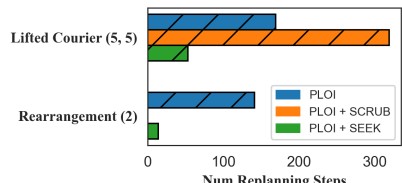

Figure 7: SCRUB on grounded domains, SEEK on lifted domains.

## 7  Concluding remarks

**Limitations.** TASKOGRAPHY currently supports only a fraction of the diverse types of planning problems possible on 3DSGs. Geared towards identifying the most promising avenues in learning-based planning, the first release of this benchmark focuses exclusively on offline task planning in fully observable and deterministic domains. Furthermore, low-level motion planning is excluded from our benchmark. Robots operating in the real world will need to reason under partial observability, sensor noise, and resource constraints.

**Outlook.** TASKOGRAPHY, in conjunction with SCRUB and SEEK aid the robot learning community by (a) providing guidelines and implementations for practitioners choosing a task planner, (b) serving as a benchmark for upcoming learning-based planners, and (c) guiding the design of futuristic spatial representations for robotic task planning. We believe TASKOGRAPHY is a first step towards addressing several of the grand challenges along the road to developing general planning capabilities for autonomous intelligent robots.

## Acknowledgements

CA and KMJ would like to thank Tom Silver and Rohan Chitnis for their help at various stages of this project, including early-stage feedback, code release, and for proofreading an initial version of this manuscript. KMJ acknowedges generous fellowship support from NVIDIA. LP acknowledges grants from IVADO and from the CIFAR Canada AI chairs program. FS acknowledges funding support from NSERC and the NFRF exploration program. The authors collectively acknowledge support during various initial stages of the project by Florian Golemo.

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
