# OpenReview forum: "Taskography: Evaluating robot task planning over large 3D scene graphs"
_robot-learning.org/CoRL/2021/Conference — CoRL2021 Poster_

### Official Review · Reviewer_TxRd · 2021-07-19

**Originality:** Good
**Technical Quality:** Very Good
**Clarity Of Presentation:** Excellent
**Impact:** 3

**Recommendation:**

Weak Accept: I recommend accepting the paper, but will not argue for my recommendation if the majority of other reviewers have a different opinion.

**Summary:**


This paper presents a large-scale robotic planning benchmark over 3DSGs that is designed to evaluate symbolic reasoning and planning over 3D scene graphs (3DSGs). The proposed benchmark consists of 20 task planning domains, including object rearrangement and courier, totaling 3734 tasks. This work benchmarks the performance of optimal planners (e.g. Fast Downward, Satplan, Delfi, etc.), satisficing planners (e.g. Fast Forward, DecStar-satisficing, Cerberus, etc.), and learning-based planners (e.g. relational policy learning, planning with learned object importance). Given the findings of benchmarking existing planning methods, two techniques have been proposed to (1) prunes a 3DSG to produce a sparsified 3DSG to improve the efficiency of classical planners and (2) ensures the reachability of all objects to reduce the number of replanning steps for learning-based planners. The experiments show the two proposed techniques yield improve performance. I believe this work presents a promising benchmark together with two effective techniques. I am leaning toward acceptance yet still have some concerns regarding how the benchmark can scale up to handle partially observable environments with stochasticity and incorporate low-level interactions, the access to the benchmark, the evaluation of the proposed techniques, and the discussion on the related work.

**Issues:**


Described in the strengths and weaknesses section.

**Reviewer Expertise:**

Good: General knowledge of the area

**Strengths And Weaknesses:**


## Paper strengths and contributions
**Motivation**
- The motivation for modeling real-world environments as 3D scene graphs is convincing.
- Utilizing 3D scene graphs modeling real-world environments to evaluate planning methods is intuitive and realistic.
- To this end, constructing a large-scale planning benchmark over 3D scene graphs is promising.

**Technical contributions**
To alleviate the issues of classical planners and learning-based planners, two techniques are proposed in this work.
- The proposed technique SCRUB prunes a 3DSG by removing vertices and edges extraneous to the task to produce a sparsified 3DSG, greatly reducing the planning time required by classical planners (e.g.  FD and FF) and enabling optimal planners to run on lifted domains.
- The proposed technique SEEK ensures that all objects are reachable from their respective room nodes to reduce the number of required replanning steps.

**Clarity**
The overall writing is very clear. The presentation is well-organized.

**Related work**
The authors did a good job explaining how this work is related to and different from other relevant prior works.

**Experimental results**
- The presentation of the experimental results is clear.
- The findings of the experimental results are well-explained, which leads to the intuition of the two proposed techniques (i.e. SCRUB and SEEK).

## Paper weaknesses and questions

**Fully observable and deterministic domains**
The current benchmark only supports the environments that are fully observable and deterministic, which is rather unrealistic to many robotics domains that require planning. I believe it would be easy to incorporate environments with stochasticity. I would like to hear the authors' opinions on how this benchmark can scale up to deal with partially observable environments.

**Low-level interactions**
Given that this benchmark does not support low-level interactions with environments, I would hardly call it a robotics environment. It seems to me that it is more suitable to call it a planning benchmark. At this point, I believe this paper oversells a little bit by saying this is a robotics benchmark.

**SCRUB and SEEK on other planning benchmarks**
While the effectiveness of SCRUB and SEEK is well-demonstrated on this benchmark, it would be even more convincing to also evaluate them on other commonly used planning benchmarks.

**Related work**
While the related work section sufficiently covers most relevant prior works, I believe it would make the discussion more comprehensive by including some works that construct and execute programs as symbolic plans, including
- Modular multitask reinforcement learning with policy sketches
- Zero-Shot Task Generalization with Multi-Task Deep Reinforcement Learning
- Programmable agents
- Program Guided Agent
- Program Synthesis Guided Reinforcement Learning

**Other domains**
While it states that 20 domains are included in this benchmark, only 2 domains are discussed in the paper. I did not find descriptions of other domains in the Supplementary material. Am I missing anything?

**The access to the API and data**
While I do recognize the difficulty to "submit" the benchmark, I still find it difficult to evaluate how good this is without actually playing with it. It would be better if more examples such as videos are shown to demonstrates how the benchmark can be used.

**Summary Of Recommendation:**


I believe this work presents a promising benchmark together with two effective techniques. I am leaning toward acceptance yet still have some concerns regarding how the benchmark can scale up to handle partially observable environments with stochasticity and incorporate low-level interactions, the access to the benchmark, the evaluation of the proposed techniques, and the discussion on the related work.

---

> ### Author Response · Authors · 2021-08-31
> **Response to review**
>
> Thank you for your comprehensive comments.  We have now added a subsection in the supplementary material, detailing our tasks; with an example for each category of tasks. In the above post titled “joint response”, we clarify how we view our task planning benchmark is valuable despite excluding low-level actions, and on the remarks about novelty/innovation.
>
> 1. **Fully observable and deterministic domains**: As you rightly note, our benchmark can be extended to incorporate partial observability / stochasticity. One aspect we should have better clarified is that our domains already incorporate partial observability in the spatial hierarchy (rooms become accessible only once a robot visits a floor, objects at a place only become accessible once a robot visits that place, and so on). We do not handle stochastic domains yet, and this is done to support a consistent planning interface across the wide range of planners we benchmark. The majority of classical planners whose heuristics require access to the full symbolic state of the graph will no longer be applicable.
> 2. **Low-level interactions**: Please refer to the post above titled “joint response” (Low-level (motion planning)). As you note, Taskography directly builds upon the Gibson scenes, allowing for robotic simulation and motion planning using a custom simulator (the Gibson or the Habitat simulation platforms). This allows future research to extend Taskography to incorporate motion planning. As stated in the “limitations” section in the paper, our current effort centers solely on task planning; as much progress needs to be made on this front alone, to enable complex real-robotic tasks.
> 3. **SCRUB and SEEK on other planning problems**: SCRUB and SEEK, by design, are only applicable to 3DSGs. This prohibits their application onto general STRIPS planning problems. A degenerate case of SCRUB that prunes neighbourhoods without regards to spatial hierarchies is the “Neighbors” baseline from PLOI (Silver et al. 2021), which was shown to be inferior to a graph neural network-based pruning method.
> 4. **Related work**: Thank you for pointers to the approaches. We added these in our related work section.
> 5. **Other domains**: Please refer to the post above titled “joint response” (clarification on tasks). We hope this clarifies issues pertaining to task specification
> 6. **Access to the benchmark**: Our benchmark will be made openly available. Gibson and the associated 3DSG data is only available under a license agreement; however all the code and planner results/logs will be made publicly available, to ensure reproducibility.
>
> Additionally, we have revised our supplementary material to contain more information about task specifications, interfaces, and hope this clarifies any remaining issues.

---

> > ### Comment · Reviewer_TxRd · 2021-09-03
> > **Re: Response to review**
> >
> > **Tasks & domains**: I do not like how the authors keep emphasizing "20 challenging robotic task planning domains" but it is actually 4 types of tasks x 5 domains. Please tone this down in the paper.
> >
> > **Partial observability & SCRUB and SEEK on other planning problems & related work**: noted and thanks.
> >
> > **Low-level interactions**: if no low-level interaction is involved, I feel selling this as a "robotics" dataset does not feel that right. It gives an impression that manipulation and locomotion are involved.
> >
> > **Access to the benchmark**: it is just really difficult to precisely judge this work without the access.
> >
> > I have read the reviewers from other reviewers and the author's response. I decided to keep my original rating.

---

### Official Review · Reviewer_aMNE · 2021-07-23

**Originality:** Fair
**Technical Quality:** Very Good
**Clarity Of Presentation:** Good
**Impact:** 2

**Recommendation:**

Weak Reject: I recommend rejecting the paper, but will not argue for my recommendation if the majority of other reviewers have a different opinion.

**Summary:**

This paper introduces a benchmark over 3D scene-graphs (3DSGs) for large-scale robotic task planning problems, along with two methods to improve existing planning and/or learning algorithms.
The proposed benchmark builds on top of an existing dataset (Gibson) and a corresponding 3DGS with a focus only on symbolic reasoning, discarding the motion planning and manipulation aspects.
Running existing algorithms on this benchmark provides insights on how they can be improved, for which two methods are proposed: sparsifying the graph by recursively pruning the original full-scale one (strategy called SCRUB), and exploiting the hierarchical graph structure to attend only on a subset of the objects present in the environment given a (importance) scoring function (strategy called SEEK).
SCRUB is shown to improve performance of planning algorithms on grounded domains, whereas SEEK provides a better performance increase on lifted domains.

**Issues:**

Please also see (Strengths & Weaknesses).

small issues:
- line 39: '... vast majority [of] the ...'
- line 202: standard deviations (detailed results) are not included in the supp. material as reported in the main text.
- line 267: missing parenthesis
- line 283: initialze
- line 294-296: contrary to the claim here, SCRUB does not improve BFWS on Courier(10,10) task according to Fig.4
- line 328: 'While SEEK, unlike SCRUB[,] is not ...'

**Reviewer Expertise:**

Very good: Comprehensive knowledge of the area

**Strengths And Weaknesses:**

- As the main strength of this work, well-structured and explained benchmarks are always appreciated as they tend to initiate faster progress in a particular domain.
- Even though the proposed benchmark, as it's presented, would mainly be useful for the planning community, if it's indeed tightly coupled (or easy to integrate) with the original benchmark environment (gibson) that it builds on, a broader robotics community would also be able to make use of it.
- The detailed analysis of different planning algorithms on a multitude of domains provides insights on the relative strengths and weaknesses of those methods, along with promising improvement directions.

weaknesses:
- the paper (including the supp. mat.) does not provide sufficient information on the tasks. I know it relies on a prior dataset, but it'd help if a subset of the tasks are explained at least in the supp. mat. to make the paper more self-contained. (The website also fails to provide further info on that point)
- Exclusion of motion and/or manipulation planning aspects diminishes relevance / usefulness of the benchmark for the robotics community. This might be alleviated if the paper explains / discusses how such low-level planning can be interfaced.
- There, usually, is a strong dependency between the motion / manipulation plans and high-level decisions, e.g., due to the executability / feasibility of a low level plan, which in turn might require re-planning, or backtracking. Also, some sequential manipulation behaviors might inform high-level decisions about their (in)feasibility. None of those are captured by a pure high-level planning-based benchmark. So, it's not clear how this benchmark might be utilized by a broader robotics research community.

**Summary Of Recommendation:**

Benchmarks play an important role in research. Traditionally, robotics field did not have many such benchmarks or datasets compared to machine learning domain. Recently, there has been a trend to introduce them for the robotics research, and this paper (and thus the introduced environments/dataset) could be a good addition. However, since motion and manipulation planning aspects are excluded, it only allows testing of high-level decision planning algorithms. Decision-making is surely a critical component of robotics research, however, interaction with the environment, thus the motion and manipulation planning and execution are the distinct aspect compared to classical AI planning. I'm not totally convinced on its usefulness in that regard, as the paper lacks proper explanations and discussion on such interfaces.
The introduced SCRUB and SEEK algorithms provide improvements over existing methods but they are rather small extensions.

---

> ### Author Response · Authors · 2021-08-31
> **Response to review**
>
> Thank you for the thorough review! We have now added a subsection in the supplementary material, detailing our tasks; with an example for each category of tasks. In the above post titled “joint response”, we clarify how we view our task planning benchmark is valuable despite excluding low-level actions, and on the remarks about novelty/innovation.
>
> 1. **Exclusion of low-level actions**: In addition to the comments listed in the “joint response” (Low-level (motion planning)), we note that our task planning benchmark directly builds upon the Gibson scenes, which allow for robotic simulation and motion planning using a custom simulator (the Gibson or the Habitat simulation platforms). This allows future research to extend Taskography to incorporate motion planning. As stated in the “limitations” section of the paper, our current effort centers solely on task planning; as much progress needs to be made on this front alone, to enable complex real-robotic tasks. The fact that we do not model the physics of contact and object interaction does not reduce the need for long term reasoning: task planning is still too slow for typical indoor environments. This effort therefore looks at task planning as the first frontier to tackle.
>
> 2. **Novelty/innovation**: Please see our post titled “joint response” above. SCRUB and SEEK uncover important shortcomings on naively applying classical task planners and modern learning-based planners on 3DSG domains. They demonstrate that, with the right sparsification method, it is possible for classical planners to perform on par with (and in cases, better than) state-of-the-art learning-based planners.
>
> We believe our open-source release will facilitate rapid benchmarking, design of newer tasks, and further work (including the incorporation of low-level motion planning).
>
> Minor remarks: Thank you for the keen eye and pointers -- all of these have been rectified in our revision, save for standard deviations / error bars; we opted against standard deviations in our tables as these applied only for runtimes for most planners (since all planners, except learning-based ones, were deterministic). Removing error bars reduced clutter and improved readability. We will open-source all our planner logs in an explorable format, which should help clarify this.

---

> > ### Comment · Reviewer_aMNE · 2021-09-01
> > **Thanks**
> >
> > Thank you for the clarifications. I still cannot justify the acceptance of this paper at CoRL. It fails to provide sufficient novelty in the learning domain, and only focuses on the high-level planning aspect within robotics (similar issues raised by other reviewers as well).  Low-level physical interaction and motion planning aspects are the distinctive complexities of the robotics domain compared to classical AI planning. As the reviewer TxRd points out, the dynamic/partially observable nature of this domain also highlights the importance of the dependency between high- and low-level planning, which is neglected within this benchmark/work.
> >
> > I'd like to emphasize that this benchmark deserves publication in a more appropriate venue, my evaluation is w.r.t. the expectations of this specific conference.

---

### Official Review · Reviewer_TvP2 · 2021-07-24

**Originality:** Fair
**Technical Quality:** Very Good
**Clarity Of Presentation:** Very Good
**Impact:** 4

**Recommendation:**

Strong Accept: I recommend accepting the paper and will argue for my recommendation even if other reviewers hold a different opinion.

**Summary:**

The work introduces a benchmark for 3d scene graphs while focusing on the complexities involved in symbolic planning. The authors further introduce two methods to alleviate common problems with planning over such 3dSGs.

**Issues:**

line 99 "lifted actions" could the authors formally define what is a "lifted" action?

I do not have any major issues with the paper since the problem domains not included in the benchmark are adressed in the Limitations section and these are totally reasonable.

**Reviewer Expertise:**

Good: General knowledge of the area

**Strengths And Weaknesses:**

- The fact that there isn't a substantial benchmark on large scale complex scene graphs is a large supporting point for this work
- I like the summarizing analyses; e.g. "Interestingly, larger scenes do not appear to directly correlate with task complexity, as the performance metrics remains largely consistent between the tiny and medium splits of the same domain (Table. 2)."

**Summary Of Recommendation:**

Overall I think the proposed work introduces an extremely useful benchmark for the community and competitive baseline augmentations to existing methods; i.e. the proposed algorithms simply take advantage of observations of deficiencies of existing works and these are well documented.

---

> ### Author Response · Authors · 2021-08-31
> **Response to review**
>
> Thank you for the positive feedback. We have revised our manuscript and supplementary material to address reviewer concerns. To answer your question, a “lifted action” is not grounded on any specific objects in the planning problem. It is rather an action schema -- a template for actions to be grounded on plannable object instances. An example each for a “lifted” and “grounded” planning problem may be found on line 104 of our manuscript.

---

### Official Review · Reviewer_PJVU · 2021-07-26

**Originality:** Good
**Technical Quality:** Good
**Clarity Of Presentation:** Very Good
**Impact:** 3

**Recommendation:**

Weak Accept: I recommend accepting the paper, but will not argue for my recommendation if the majority of other reviewers have a different opinion.

**Summary:**

3D scene graph is an unified and informative representation of the scene. Many existing works utilize 3D scene graph to solve tasks that require long horizonal planning and reasoning. While previous works directly take the complete scene graph as input, this work proposes a planner-agnostic algorithm to prunes the scene graph in linear time based on the goal of the task, and further propose a procedure for learning based planner to ensure the connectivity of the pruned subgraph.

**Issues:**

Figure 4 in the paper is ambiguous since the ellipses of different methods are overlapped with each other, it should be re-drawn.

**Reviewer Expertise:**

Very good: Comprehensive knowledge of the area

**Strengths And Weaknesses:**

Strengths:
1.	The idea of pruning the large 3D scene graph and utilize the subgraph to solve tasks is convincing.
2.	The designed pruning algorithm SCRUB which builds the subgraph iteratively is effective since it could finish in linear time.
3.	Successive experiments have been implemented to verify the effectiveness of the proposed algorithm.
Weaknesses:
1.	The proposed pruning algorithm SCRUB is lack of innovation. I think it is similar with BFS and just directly merge all nodes and edges between robot and goal objects into the subgraph. In complex scenes and tasks, the algorithm might fail to prune enough nodes and edges to improve the efficiency.
2.	This workload of this paper with respect to robot learning is not sufficient.  It could introduce learning methods at the pruning algorithm or design a learning-based planner that fits the pruning algorithm.
3.	In real world application, the 3D scene graph always have some incorrect nodes and edges. Since the pruning algorithm heavily depends on the accuracy of the scene graph, I wonder whether the planner could work well with the pruned graph in physical world. While this work does not verify the robustness of the algorithm on the imperfect 3D scene graph.


**Summary Of Recommendation:**

This work makes contribution to the application of 3D scene graph. Pruning 3D scene graph to improve the performance and efficiency of the planner is a novel and worthy of in-depth study direction. Meanwhile, this work provides a large scale benchmark to support further research and evaluation. The successive experiments illustrate that the pruning algorithm could work with both non-learning and learning-based planners well. However, the proposed algorithm has its limitation in terms of generalization and robustness.

---

> ### Author Response · Authors · 2021-08-31
> **Response to review**
>
> We acknowledge that real-world 3D scene graphs are prone to noisy node and edge attributes (or even the existence thereof). We have consciously not ventured into this area since (a) we intend the focus of taskography to be on task planning -- a common assumption in robot task planning is that of near-perfect perception; this enables us to fairly benchmark planners without regards to errors introduced by perception modules; (b) 3DSGs are a nascent, active area of research -- since we do not yet know the exact kinds of errors that manifest when constructing real-world 3DSGs, it will be not particularly effective to introduce a ‘synthetic’ noise model; more so since we anticipate for Taskography to be a large-scale benchmark for 3DSG planning.
>
> An aside to the comment “Many existing works utilize 3D scene graphs to solve long horizon planning and reasoning” --- to the best of our knowledge, 3DSGs have not thus far been employed in long-horizon task planning (with the exception of object search [A]). We believe that Taskography is unique in its scale and positioning, intersecting long-horizon robot task planning and 3DSGs and offering researchers a path to custom domain creation and rapid experimentation.
>
> Responses to weaknesses in order:
>
> 4. We argue that SCRUB is effective despite its simplicity. The vast majority of “complex” household robotics tasks can be categorized as tasks that: (a) involve many objects in a few rooms (e.g. clean room, clean kitchen); (b) involve few objects in many rooms (e.g. collect / rearrange these objects around the house), or (c) many objects in many rooms. Moreover, 3DSGs are hierarchical structures, where the lower layers constitute the majority of nodes (objects / receptacles). SCRUB efficiently sparsifies (linear in nodes + edges) the 3DSG, and should the task at hand fall into categories (a) or (b), SCRUB will prune all but few nodes at the lower layers of the tree. *Because plan time savings are exponential with respect to the number of nodes pruned, as shown by Silver and Chitnis et al in PLOI, SCRUB remains effective for grounded planning problems despite task or scene graph complexity and always yields a minimal and valid state-space.*
>
> The caveat here is a task that involves many objects across many rooms. However, it's very difficult to conceptualize realistic household tasks that fit this description, and if one exists, we argue that such a task would be separable into subtasks of type (a) or (b) in which case the most fitting of our proposed methods can be applied.
>
> Despite this simplicity, SCRUB greatly improves the performance of classical planners, a surprising finding.
>
> 5. Effective domain-agnostic learning-based planners for state-space pruning already exist (cf. PLOI). We demonstrate the effectiveness of this algorithm across all domains, and propose to couple it with SEEK which drastically robustifies the validity of inferred subgraphs by ensuring all important nodes are reachable. The PLOI + SEEK of Hierarchical PLOI + SEEK variant seems to fit this description exactly.
>
> 6. All our domains are built upon Gibson’s Tiny and Medium 3DSG splits. While the Tiny split consists of human verified scenes, the Medium split comprises far larger 3DSGs that have been automatically generated from ego-centric observations without human post-processing. We argue that such 3DSGs reflect some of the imperfections that one would encounter when building these structured representations in the real-world.
>
> We are not sure how to interpret the following remark: “the proposed algorithm has its limitation in terms of generalization and robustness”. Both SCRUB and SEEK have been applied without any sort of hyperparameter tuning, across all domains in the benchmark. Our results demonstrate that SCRUB and SEEK are highly performant, and we believe this to be compelling evidence for generalization.
>
> **Referenced papers**:
>
> [A] Semantic and Geometric Modeling with Neural Message Passing in 3D Scene Graphs for Hierarchical Mechanical Search. ICRA 2021.

---

### Author Response · Authors · 2021-08-31
**Joint response**

We thank the reviewers for their comprehensive evaluations of the work. All reviewers recognized the importance and value of such a benchmark, and liked the “detailed analyses” and “insights” that ensued.

We first respond to three smaller concerns listed in the meta review, and where applicable, explain how these have been addressed in the revision.

1. **Clarification on tasks**: Reviewers `aMNE` and `TxRd` sought further information about the tasks. Taskography includes 4 categories of tasks (rearrangement, courier, and their lifted variants). Each task category is instantiated in 5 unique domains, totalling to 20 domains across all tasks combined. Our supplementary material now includes a section (1.2) introducing the tasks, with an example for each. We hope that our open-source release will help in the dissemination and reproducibility of all nuances associated with the tasks in the benchmark.

2. **Low-level (motion planning)**: Reviewers `aMNE` and `TxRD` sought clarity on the applicability of our benchmark to robotic tasks. Taskography does not explicitly consider motion planning, which imposes additional kinodynamic constraints on the planning problem. We state this design choice upfront in our manuscript. We believe that the findings from Taskography are very useful to robotics researchers and practitioners even without incorporating motion planning constraints.
* (a) A dominant paradigm to task and motion planning (TAMP), one that is still used in several modern day robotics stacks is to decouple task planning from motion planning (i.e., computing a motion plan after a task plan has successfully been found). While the TAMP literature has argued for tighter coupling between the task and the motion planners, decoupled task and motion planning has advantages in terms of simplicity and explicability. Such decoupled ‘task-then-motion’ planning stacks require the task planners to be very efficient (as they potentially have to be invoked multiple times until a viable motion plan may be computed). The Taskography benchmark, and SCRUB and SEEK, present practitioners with real-time and efficient task planners for use with motion planners.
* (b) Different from other symbolic benchmarks prevalent in the AI planning community, we ground our planning problems in contemporary robotics problems; ones that are actively being tackled by the robotics, computer vision, and learning communities. Our primary tasks are consistent with the notion of  “rearrangement” planning -- as outlined in a recent multi-institutional whitepaper [A]; widely being pursued as the next frontier for embodied AI applications. Furthermore, all of our tasks at least involve coarse motion planning (gridworld-like, without kinodynamic constraints), which is not commonplace in existing AI planning benchmarks. AI planning benchmarks, on the other hand, do not require explicit factoring in of such spatial hierarchies and constraints. We require our planners to explicitly compute paths between interactable scene entities, without relying on an oracle “goto” action, taking this one step closer to robotic tasks. In several real robot software stacks, a “high-level” planner (such as A*, D* lite, or similar graph planner) produces a coarse plan which is then refined by a “low-level” planner (Bi RRT*, etc.). Path plans produced by planners on the Taskography benchmark tasks can be similarly construed as “high-level” plans, which are ready for consumption by “low-level” planners.

3. **Novelty/innovation**: We emphasize that the taskography benchmark unveils multiple surprising / counterintuitive findings to influence research in both the 3DSG and planning communities. First, we show that --- contrary to claims in [B, C, D] --- 3DSGs are not as easily amenable to downstream task planning (except for simpler, object search settings). Second, we demonstrate pruning as an effective and tractable mechanism for planning over 3DSGs (our proposed technique, SCRUB, prunes a 3DSG and enables classical planners to perform on par with state-of-the-art learning-based planners). These insights could be helpful in the design of the next generation of more performant task planners over 3DSGs, and help both robotics researchers and practitioners.

**Referenced papers**:

[A] Rearrangement: A Challenge for Embodied AI. arXiv 2020.

[B] 3D Dynamic Scene Graphs: Actionable Spatial Perception with Places, Objects, and Humans. RSS 2020.

[C] Semantic and Geometric Modeling with Neural Message Passing in 3D Scene Graphs for Hierarchical Mechanical Search. ICRA 2021

[D] 3D Scene Graph: A Structure for Unified Semantics, 3D Geometry and Camera. ICCV 2019.

---

### Meta-Review · Area_Chair_2etH · 2021-08-13

**Recommendation:** Accept (Poster)
**Confidence:** 5

**Metareview:**

In this paper, a large-scale robotic task planning benchmark over scene graphs is established. A task-conditioned 3DSG parsification method called SCRUB, and SEEK, a procedure enabling learning-based planners to exploit 3DSG structure is proposed. All reviewers recognized the importance of the benchmark to the community. While there are some concerns about the exclusion of low-level motions, which are also considered important in robotic planning tasks. And some more detailed introduction of other tasks is lacking.

In the rebuttal session, the authors have responded to the reviewers' concerns and more details are provided. All reviewers recognize the value of the benchmark to the robot learning community, and have reached a consensus of accepting this paper.

---

### Decision · Program_Chairs · 2021-09-13

**Decision:**

Accept (Poster)

**Comment:**

In this paper, a large-scale robotic task planning benchmark over scene graphs is established. A task-conditioned 3DSG parsification method called SCRUB, and SEEK, a procedure enabling learning-based planners to exploit 3DSG structure is proposed. All reviewers recognized the importance of the benchmark to the community. While there are some concerns about the exclusion of low-level motions, which are also considered important in robotic planning tasks. And some more detailed introduction of other tasks is lacking.

In the rebuttal session, the authors have responded to the reviewers' concerns and more details are provided. All reviewers recognize the value of the benchmark to the robot learning community, and have reached a consensus of accepting this paper.